# The Effect of Morpholine on Composite-to-Composite Repair Strength Contaminated with Saliva

**DOI:** 10.3390/polym14214718

**Published:** 2022-11-04

**Authors:** Awiruth Klaisiri, Siriwan Suebnukarn, Nantawan Krajangta, Thanasak Rakmanee, Tool Sriamporn, Niyom Thamrongananskul

**Affiliations:** 1Division of Restorative Dentistry, Faculty of Dentistry, Thammasat University, Pathum Thani 12120, Thailand; 2Division of Endodontics, Faculty of Dentistry, Thammasat University, Pathum Thani 12120, Thailand; 3Thammasat University Research Unit in Restorative and Esthetic Dentistry, Thammasat University, Pathum Thani 12120, Thailand; 4Department of Prosthodontics, College of Dental Medicine, Rangsit University, Pathum Thani 12000, Thailand; 5Department of Prosthodontics, Faculty of Dentistry, Chulalongkorn University, Bangkok 10330, Thailand

**Keywords:** bond strength, morpholine, repair, resin composite, surface treatment

## Abstract

The aim of this study was to specifically explore the effects of morpholine on chemical surface treatments of aged resin composites contaminated with saliva to new resin composite repair strength. One hundred and thirty five resin composite specimens were fabricated and thermocycled to replicate an aged resin composite. These aged resin composites were randomly separated into nine groups (n = 15) depending on the various surface contaminants and surface treatment techniques. These groups were as follows: group 1—no surface treatment; group 2—no saliva + adhesive agent; group 3—no saliva + morpholine + adhesive agent; group 4—no saliva + morpholine; group 5—saliva; group 6—saliva + adhesive agent; group 7—saliva + morpholine + adhesive agent; group 8—saliva + morpholine; and group 9—saliva + phosphoric acid + adhesive agent. A mold was covered on the top of the specimen center and then filled with resin composite. The shear bond strengths and failure modes were examined. The collected data was analyzed using one-way ANOVA, and the significance level was determined using Tukey’s test. Group 5 (3.31 ± 0.95 MPa) and group 6 (4.05 ± 0.93 MPa) showed the lowest bond strength statistically, while group 3 (23.66 ± 1.35 MPa) and group 7 (22.88 ± 1.96 MPa) showed the most significantly high bond strength. The bond strength in group 2 (16.41 ± 1.22 MPa) was significantly different from that in group 1 (9.83 ± 1.13 MPa), group 4 (10.71 ± 0.81 MPa), and group 8 (10.36 ± 1.53 MPa), while group 9’s (17.31 ± 1.48 MPa) SBS was not significantly different. In conclusion, the application of morpholine on aged resin composite with or without contamination with saliva prior to the application of the adhesive agent increased the bond strength of aged resin composite repaired with new resin composite (*p* < 0.05).

## 1. Introduction

Over the past few decades, the application of resin composite material for filling both anterior and posterior teeth has continued to rise due to its potential to closely match the color of the natural teeth, to patient concern about esthetics, and to the natural tooth structure being conservative. Resin composite filling materials have seen significant progress in terms of material strength, handling qualities, esthetic aspects, and durability [1]. All restorations may eventually fall out owing to secondary caries, microleakage, discoloration, and fracture, but resin composite may have the added benefit of being easily repaired rather than needing to be replaced totally [2]. One of the advantages of choosing resin composite fillings is that they are repairable and serve the current minimal invasive dentistry concept [3].

In order to accomplish resin composite repairs, obtaining excellent bonding to the existing restoration is a crucial goal [4]. The remaining resin composite filling could have remained for a lengthy and frequently unknowable amount of time in the oral environment. Adhesion to old resin composite restorations is very challenging; the oxygen inhibition layer is removed as a result of aging and saliva/water absorption, and unsaturated double carbon–carbon bonds are reduced [5]. Considering this, to increase the resin composite materials’ ability to establish strong bonds during repairs, surface treatment is applied to the aged resin composite that serves as the bonding surface [6]. Different methods of surface modification have been developed for this aim based on mechanical and/or chemical adhesion principles, although it is unclear whether treatment improves the effectiveness of restoration. By roughening the aged resin composite using techniques like air abrasion and bur abrasion, the mechanical bonding is increased, whereas the chemical adhesion is increased by applying substances like adhesive agent to the aged resin composite [7,8]. Sismanoglu reported that, under clinical situations, the practical option would be to apply adhesive agent with previous acid etching to produce an adequate repair efficacy [9].

Morpholine is a heterocyclic organic chemical compound. This heterocycle has functional groups for amine and ether. Morpholine is a base because it contains an amine [10]. Morpholine, which appears in various drugs and bioactive molecules, is a structure of considerable interest in medical chemistry [11]. In the synthesis of organic compounds, morpholine is frequently employed. Morpholine is frequently used in industry as a solvent for chemical processes because of its low cost and polarity. However, morpholine is not used in dentistry.

Strong evidence exists that the use of adhesive agents in the repair technique increases the adhesion between aged and new resin composites in vitro [12,13]. Although it has been recommended that various chemical agents be used as pretreatments to enhance the bonding of aged resin composites to one another, there has been no success in determining the most effective technique, which is essential for the repair procedure [14]. In this research, we are specifically exploring the effects of morpholine on chemical surface treatments of aged resin composites contaminated with saliva to new resin composite repair strength. The null hypothesis was that morpholine has no impact on the repair bond ability of aged resin composite contaminated with saliva to new resin composite.

## 2. Materials and Methods

The Human Research Ethics Committee of the Faculty of Dentistry, Chulalongkorn University (HREC-DCU), research code number 2022-003, approved the experimental portion of this research as ethically acceptable.

### 2.1. Preparation of Aged Resin Composite

Using the G*Power 3.1 program, the sample size was determined using 0.05 as the significance level and 0.95 as the power. A sample size of 15 specimens was utilized in each group, according to the findings of the pilot testing. One hundred and thirty five resin composite specimens (Filtex Z350 XT (A1E), 3M ESPE, St. Paul, MN, USA) were fabricated by placing resin composites in a 6 × 6 × 4 mm stainless steel mold. Two incremental layers of resin composite of 2 mm each were light-activated for 40 s (Elipar Freelight2, 3M ESPE, St. Paul, MN, USA). A celluloid strip was placed upon the top surface to make a flat surface. A sample was taken from the stainless steel mold. A second 40 s light cure was conducted either side of the resin composite. A thermocycler machine (Erios, Sao Paulo, SP, Brazil) thermocycled the resin composite samples 5000 times at a dwell length of 5 s at 5 °C and 55 °C to replicate an aged resin composite. To assess resin composite surface imperfections, every specimen was examined at a magnification of 40× on a stereo microscope. (SZ61, Olympus Corporation, Tokyo, Japan). The faulty specimens were not included.

The aged resin composites were placed in a poly(vinylchloride) tube with dental gypsum type IV in the center (Figure 1). Wet polishing with silicon carbide abrasive sheets of 320- and 600-grit was used to refine the specimens’ top surfaces (3M Wetordry abrasive sheet, 3M ESPE, St. Paul, MN, USA). On all aged resin composite specimens were performed a 10-min ultrasonic-cleaning procedure in distilled water, followed by a 10 s triple-syringe air-drying procedure.

### 2.2. Surface Contamination and Surface Treatment Protocols

The materials used in this experiment are presented in Table 1. The samples were randomly separated into 9 groups (n = 15) depending on the various surface contaminants and surface treatment techniques. The samples were handled as follows:

Group 1 (control): no surface treatment.

Group 2: no saliva + adhesive agent (SB).

Group 3: no saliva + morpholine (MOR) + adhesive agent.

Group 4: no saliva + morpholine.

Group 5: saliva.

Group 6: saliva + adhesive agent.

Group 7: saliva + morpholine + adhesive agent.

Group 8: saliva + morpholine.

Group 9: (positive control): saliva + phosphoric acid (PHOS) + adhesive agent.

### 2.3. Saliva Contamination

A single healthy person’s (no medical conditions and non-medications) unstimulated saliva was taken at the site and at the same time. It must be taken at least an hour after eating or drinking anything, as well as right before the saliva is prepared. A microbrush (Kerr Corporation, Orange, CA, USA) was used to apply 100 microliters of saliva to the aged resin composite, and it was then carefully dried for 20 s using triple-syringe air.

### 2.4. Phosphoric Acid Etch

A 37% phosphoric acid (Dentalife, Ringwood, Australia) was used to etch the specimen for 15 s. After cleaning it with water, it was dried for 10 s with triple-syringe air.

### 2.5. Morpholine Surface Treatment

9.8% morpholine (Loba Chemie PVT Ltd., Mumbai, India) was prepared by diluting 98% morpholine in distilled water, 1 mL to 100 mL, respectively. A disposable micro applicator was used to apply a thin coating of 10 microliters of 9.8% morphine from the micropipette, which was then gently triple-syringe blown for 20 s.

### 2.6. Adhesive Agent Treatment

Applying a Scotchbond multipurpose adhesive (3M ESPE, St. Paul, MN, USA) with a reusable micro applicator, the specimen was then carefully dried for 20 s with triple-syringe air. Following that, the specimen was activated by light for 20 s.

### 2.7. Bonding Procedures

Scotch blue Painter’s tape (3M, Maplewood, MN, USA) is 80 μm thick and was cut into a 6 × 6 mm square. Using a hole-puncher, a hole 2.38 mm in diameter was punched out of the adhesive tape’s center to indicate the bonding region. On the surface of each specimen, the adhesive tape was placed and then an ultradent mold (2.38 mm in diameter and 2 mm in height) was applied onto the hole of adhesive tape. The resin composite (A4B shade) was pushed against the aged resin composite before being light-cured for 40 s. Before shear bond testing, all bonded samples were preserved for 1 day in distilled water at 37 °C.

### 2.8. Shear Bond Strength (SBS) Test

The specimen was set up on the universal testing equipment (EZ-S 500N, Shimadzu Corporation, Kyoto, Japan). A shearing blade with a notched edge was used to test the SBS (Figure 2). The notched-edged shearing blade was positioned close to the bonding region. The crosshead moved at a pace of 1 mm per minute. The bond strength was computed in megapascal (MPa) by dividing the highest load (N) before the bond breakdown by the bonding region (mm^2^) [15,16,17] between the fresh resin composite and the aged resin composite.

### 2.9. Failure Mode Analysis

The failure mode was modified from Matinlinna et al. [18] and Klaisiri et al. [19] and was categorized into three types: (1) Less than 40% of the new resin composite (or old resin composite) can be seen on the surface of the old resin composite (or new resin composite), at which point an adhesive failure has occurred. (2) When at least 60% of the new resin composite (or old resin composite) is visible on the old resin composite (or new resin composite) surface, a cohesive failure has occurred. (3) When more than 40% but less than 60% of the new resin composite (or old resin composite) is visible on the surface of the old resin composite (or new resin composite), a mixed failure has occurred.

The representative samples from each group were randomly selected and analyzed to determine the failure mode. A scanning electron microscope (SEM, FEI company, Versa 3D, Eugene, OR, USA) was used to examine the samples at a 250× magnification to analyze the failure mode.

### 2.10. Statistics Analysis

The statistical study was conducted using the Windows 22.0 version of IBM SPSS statistics (SPSS Inc., Chicago, IL, USA). A one-way ANOVA was used to compute the effect of different chemical surface treatment methods on the aged resin composite to new resin composite repair strength in terms of SBS value, followed by a Tukey’s post hoc test to analyze all possible pairwise comparisons. The statistical analysis was conducted using an alpha value of 0.05.

## 3. Results

In the present investigation, Table 2 indicates the SBSs’ means and standard deviations. Group 5 and group 6 showed statistically lowest SBSs, while group 3 and group 7 showed significantly highest SBSs.

One-way ANOVA determined that the SBS in group 2 was significantly different from that in group 1, group 4 and group 8 (*p* < 0.05), while group 9’s SBS was not significantly different (*p* > 0.05).

The distributions of fracture modes are reported in Table 2. Adhesive was the most common failure mode in all groups.

The SEM photographs of the adhesive failure mode in groups 1, 4, 5, 6, and 8 are shown in Figure 3. The SEM photographs of the mixed failure mode in groups 2, 3, 7, and 9 are shown in Figure 4.

## 4. Discussion

This research investigated the effects of morpholine on chemical surface treatments of aged resin composites contaminated with saliva to new resin composite repair strength. The null hypothesis was that morpholine has no impact on the repair bond ability of aged resin composite contaminated with saliva to new resin composite. The result found that the SBS from the morpholine groups was significantly different. The null hypothesis was therefore disproved.

One of the significant issues affecting the durability of adhesive restorations is aging. During the aging mechanism, hydrolysis occurs when water molecules enter the resin matrix of the material [20]. In this study, thermocycling 5000 times at 5 °C and 55 °C was used to model the aging mechanism of resin composites in clinical settings. It has been suggested that around 5000 cycles might replicate a service period of six months in an oral environment [21]. Thermocycling creates stresses as a result of variations in the thermal expansions of distinct materials, which might cause bond breakdown at the matrix–filler interface of resin composites [22]. Rinastiti et al. showed that aging results in created surface roughness and decreased repair bond ability [23]. The aged resin composite surface treatment serves two purposes: (i) to eliminate the outermost layer affected by the saliva and reveal a fresh, more efficient surface of resin composite for wettability and (ii) to create rough surface in order to enhance surface area [24]. Dentists frequently use a bur to abrade the old resin composite surface before restoring it with new resin composite. Therefore, in this investigation 320- and 600-grit abrasive papers were used to make the old resin composite surfaces rough.

One of the big concerns with adhesive techniques that has been reported in the literature is saliva contamination [25,26]. An acid-etched tooth/restorative surface assimilates salivary components quickly and lowers surface energy, making it unsuitable for bonding [27]. Saliva contamination is a concern when new increments of resin composite should be improved, as the longer time necessary for insertion and resin composite polymerization might make contamination management more challenging. Once the rubber dam is removed, the contour of the restoration is repaired. It might also be a concern whenever using a dental rubber dam is not possible during lengthy clinical procedures.

Morpholine is a simple heterocyclic secondary amine with a wide variety of uses and industrial relevance. Tetrahydro-1,4-oxazine, tetrahydro-2H-I,4-oxazine, diethylene oximide, and diethyleneimide oxide are all names for morpholine [11]. Morpholine is widely utilized in medicinal chemistry and industry because of its favorable biological, physicochemical, and metabolic characteristics, as well as its relatively simple synthesis pathways. Morpholine application can be categorized into (i) non-cosmetic, including use as an antioxidant, corrosion inhibitor, or wax and polish emulsifier as a component of vegetable- and fruit-protection coatings and (ii) cosmetic categories, such as mascara [28]. In this study, it was found that the aged resin composite contaminated with saliva had reduced SBS when using an adhesive agent, while the groups with morpholine treatment had no effect on the saliva contaminated. Moreover, when application of morpholine on the aged resin composite contaminated with saliva occurred prior to the adhesive agent, it was found that the SBS was higher than in the positive control group, which had phosphoric etching before the adhesive agent. Yin et al. [29] reported that when oral fluids like saliva and blood contaminate the resin composite, using phosphoric acid etching may be suggested to remove the contaminate. The possible functions of morpholine may have two mechanisms. First, morpholine could decontaminate saliva on an aged resin composite to remove debris and clean the surface. Second, the matrix of an aged resin composite may be partially dissolved in morpholine, which creates a micromechanical roughness to prepare for the infiltration of resin monomers. As a result of these mechanisms, the SBS values of groups 3 and 7 were significantly higher than those of any other group. This investigation concluded that the SBS of old resin composites repaired with new resin composites was enhanced by applying morpholine to them before applying the adhesive agent, whether there was contamination from saliva or not.

Regarding the failure mode of this research, mainly adhesive failures were seen in all of the groups. As demonstrated by groups 2, 3, 7, and 9, high SBS was frequently associated with mixed failures [15,17,30]. Cohesive failures were not seen in this investigation.

The study’s limitations: First, the specimens were incubated for only 24 h. Thermocycling for the durability of bonding was not performed. Second, the clinical effectiveness of an adhesion mechanism is affected by a variety of factors, not only the SBS. As a result, it is important to interpret the outcomes of our research carefully.

## 5. Conclusions

Within the limitations of this research, we are specifically exploring the effects of morpholine on chemical surface treatments of aged resin composites contaminated with saliva to new resin composite repair strength. We found that the application of morpholine on aged resin composite with or without saliva contamination prior to the adhesive agent increased the SBS value (22.88 ± 1.96, 23.66 ± 1.35 MPa, respectively) of aged resin composite repaired with new resin composite compared to the positive control (17.31 ± 1.48 MPa). An alternative clinical approach for resin composite repair uses morpholine to treat the surface of aged composites before applying the adhesive agent.

## Figures and Tables

**Figure 1 polymers-14-04718-f001:**
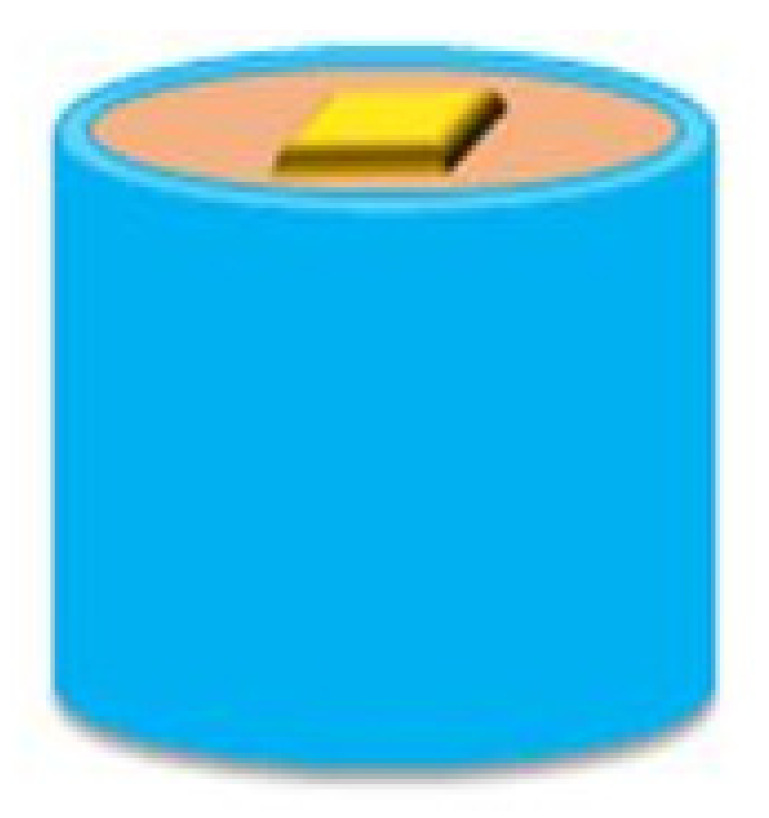
Illustration of specimen embedded in the poly(vinylchloride) tube.

**Figure 2 polymers-14-04718-f002:**
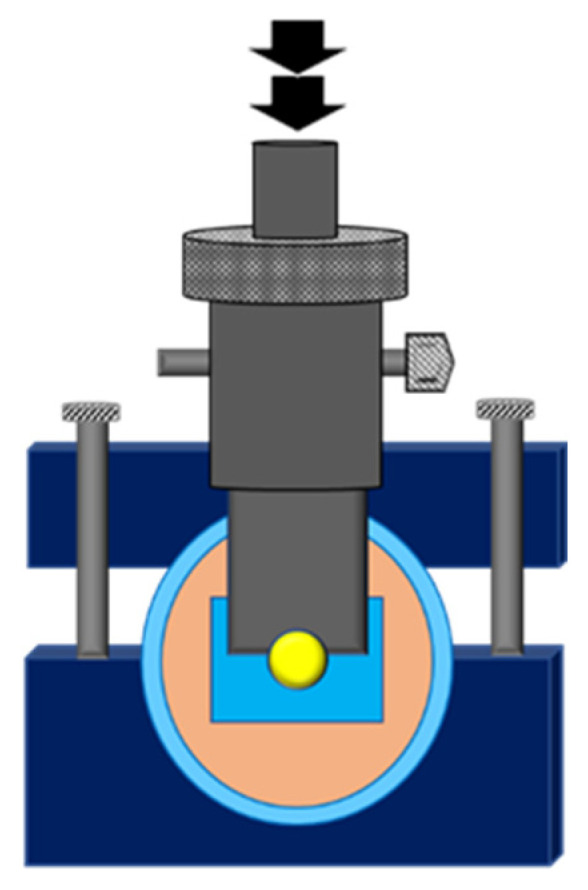
Illustration of notched-edge shear bond strength test.

**Figure 3 polymers-14-04718-f003:**
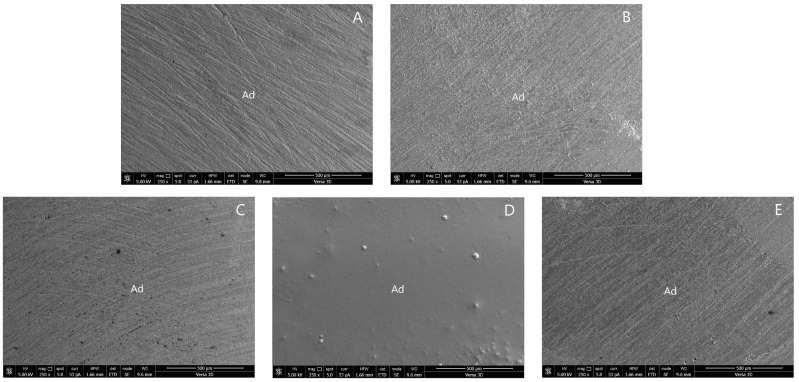
SEM photographs of adhesive failure mode: (**A**) group 1; (**B**) group 4; (**C**) group 5; (**D**) group 6; and (**E**) group 8. (Ad, adhesive failure).

**Figure 4 polymers-14-04718-f004:**
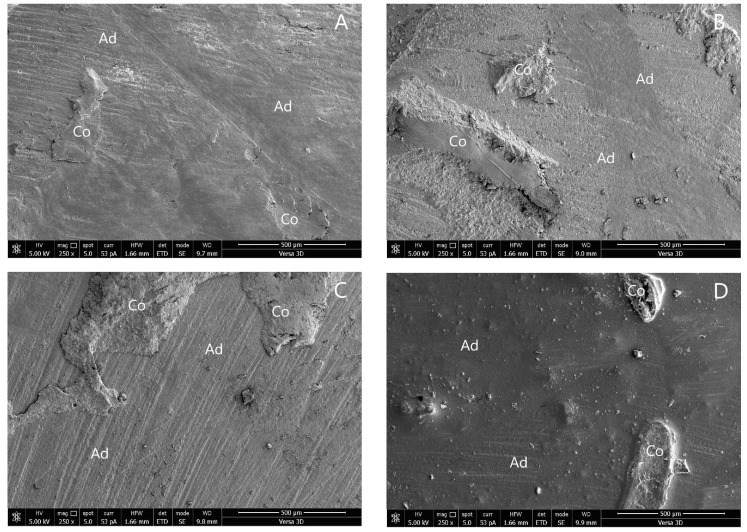
SEM photographs of mixed failure mode: (**A**) group 2; (**B**) group 3; (**C**) group 7; and (**D**) group 9. (Ad, adhesive failure; Co, cohesive failure in new resin composite).

**Table 1 polymers-14-04718-t001:** The materials used in this investigation.

Type	Material	Composition
Filling material	Resin composite (Filtex Z350 XT (A1E, A4B), 3M ESPE, St. Paul, MN, USA)	Bis-GMA, Bis-EMA-6, TEGDMA, UDMA, PEGDMA, silane-treated ceramic, silane-treated silica, silane-treated zirconia
Adhesive agent	Scotchbond multipurpose adhesive (3M ESPE, St. Paul, MN, USA)	Bis-GMA, HEMA, peroxide component of catalyst resin, amine
Surface treatment agent	Morpholine (Loba Chemie PVT Ltd., Mumbai, India)	98% Extra pure O(CH_2_CH_2_)_2_NH

Abbreviations: Bis-GMA, bisphenol A-glycidyl methacrylate; Bis-EMA-6, bisphenol A polyethylene glycol diether dimethacrylate; TEGDMA, triethylene glycol dimethacrylate; UDMA, urethane dimethacrylate; PEGDMA, polyethylene glycol dimethacrylate; HEMA, 2-hydroxyethyl methacrylate.

**Table 2 polymers-14-04718-t002:** The SBSs’ means and standard deviation (MPa) and failure mode percentage.

Group	Mean SBS ± SD		Failure Mode	Cohesive
		Adhesive	Mixed	
1. No surface treatment	9.83 ± 1.13 ^a^	100	0	0
2. No saliva + SB	16.41 ± 1.22 ^b^	80	20	0
3. No saliva + MOR + SB	23.66 ± 1.35 ^c^	60	40	0
4. No saliva + MOR	10.71 ± 0.81 ^a^	100	0	0
5. Saliva	3.31 ± 0.95 ^d^	100	0	0
6. Saliva + SB	4.05 ± 0.93 ^d^	100	0	0
7. Saliva + MOR + SB	22.88 ± 1.96 ^c^	70	30	0
8. Saliva + MOR	10.36 ± 1.53 ^a^	100	0	0
9. Saliva + PHOS + SB	17.31 ± 1.48 ^b^	80	20	0

The value with identical letters indicates no statistically significant difference.

## Data Availability

Not applicable.

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
