# Peer review of "The Effect of Morpholine on Composite-to-Composite Repair Strength Contaminated with Saliva"

_polymers, 2022, doi:10.3390/polym14214718_

Round 1
Reviewer 1 Report
Abstract: The first sentence of this section should begin with "the aim of this study". There is a lack of meaning in the sentence.
The p value is not mentioned in the conclusion part. This should be added.
Introduction: It’s well written.
Materials and methods: How were the sample size specified in lines 79 and 80 determined? Please specify if an ISO standard was used. Or cite if there is a related publication.
In lines 84-85, light-curing of composite resins for 40 seconds is mentioned. However, the manufacturer stated in the instructions for use those 20 seconds is sufficient. Why was light cured for 40 seconds when 20 seconds was sufficient?
Long names of abbreviations should be indicated in Table 1. Like Bis-GMA...
The type of material used in Table 1 should also be specified in a separate column.
What are the inclusion and exclusion criteria of the person whose saliva is used mentioned in line 120? State it clearly.
Results: The data packet in the result outputs is very limited. For example, it increases the value of working with data such as SEM data.
Discussion: Please mention more of the limitations of the study.
References: References 3, 10, 21, 22, 24, 25 and 27 are very old. Please revise with new ones.
Reviewer 2 Report
In general, the manuscript seems to be very concise, but to the point and interesting. Some revisions which are suggested are as follows:
1) The abstract of the paper needs to be re-written. Instead of listing all groups' names it will be better to describe them in few sentences pointing out the differences between them. Additionally, abstract should contain some quantified data.
2) First paragraph of section Introduction should be supplemented with some more sentences presenting some examples of resin composite filling materials.
3) Caption of Figure 1: the notation of polymer name should be corrected; it should be "poly(vinyl chloride)" instead of "polyvinylchloride".
4) Caption of Table 1. needs to be changed.
5) Final conclusions should be extended, supplemented with some quantified data and with the perspectives.
Round 2
Reviewer 1 Report
Thank you to the authors for making the corrections I suggested. As such, it is acceptable.